# Sleeve Gastrectomy and Roux-En-Y Gastric Bypass. Two Sculptors of the Pancreatic Islet

**DOI:** 10.3390/jcm10184217

**Published:** 2021-09-17

**Authors:** Gonzalo-Martín Pérez-Arana, José Fernández-Vivero, Alonso Camacho-Ramírez, Alfredo Díaz Gómez, José Bancalero de los Reyes, Antonio Ribelles-García, David Almorza-Gomar, Carmen Carrasco-Molinillo, José-Arturo Prada-Oliveira

**Affiliations:** 1Department of Human Anatomy and Embryology, Faculty of Medicine, University of Cadiz, 11003 Cadiz, Spain; jose.vivero@uca.es (J.F.-V.); alonso.camacho@uca.es (A.C.-R.); Antonio.ribelles@uca.es (A.R.-G.); carmen.carrasco@uca.es (C.C.-M.); 2Institute for Biomedical Science Research and Innovation (INIBICA), University of Cadiz, 11003 Cadiz, Spain; david.almorza@uca.es; 3Asociación Gaditana de Apoyo al Investigador (AGAI), 11012 Cadiz, Spain; 4Surgery Unit, Puerta del Mar Universitary Hospital, University of Cadiz, 11003 Cadiz, Spain; 5San Carlos Hospital, Andalusian Health System, 28040 Madrid, Spain; diazgomez.a@gmail.com; 6Badajoz Hospital, Extremadura Health System, 06001 Badajoz, Spain; josebancalerodelosreyes@gmail.com; 7Operative Statistic and Research Department, University of Cadiz, 11003 Cadiz, Spain

**Keywords:** sleeve gastrectomy, roux-en-Y gastric bypass, beta-cell, alpha-cell, epsilon-cell, islet, trans-differentiation

## Abstract

Several surgical procedures are performed for the treatment of obesity. A main outcome of these procedures is the improvement of type 2 diabetes mellitus. Trying to explain this, gastrointestinal hormone levels and their effect on organs involved in carbohydrate metabolism, such as liver, gut, muscle or fat, have been studied intensively after bariatric surgery. These effects on endocrine-cell populations in the pancreas have been less well studied. We gathered the existing data on these pancreatic-cell populations after the two most common types of bariatric surgery, the sleeve gastrectomy (SG) and the roux-en-Y gastric bypass (RYGB), with the aim to explain the pathophysiological mechanisms underlying these surgeries and to improve their outcome.

## 1. Introduction

Bariatric/metabolic surgery has been a powerful tool for the treatment of diabetes mellitus for a long time. Sleeve gastrectomy (SG) and roux-en-Y gastric bypass (RYGB) are two of the most performed ones [1,2] as Figure 1 shows.

Changes in energy homeostasis and body fat mass have been proposed as a primary mechanism to explain these phenomena [3,4], but other mechanisms such as changes in several gastrointestinal hormones also seem to be involved with a large number of publications written on the topic. Many of them have related the anatomical changes in the gastrointestinal tract after surgery with the modification of serum levels of glucagon like peptide-1 (GLP-1) [5], ghrelin [6], peptide tyrosine-tyrosine (PYY) [7], gastrointestinal inhibitory peptide (GIP) [8], or even leptin [9], among others, in humans and animal models. Their involvement is clear, but the exact mechanisms and their degree of participation remain partially unknown.

At the other end of the entero-pancreatic axis, the endocrine pancreas containing Langerhans islets determines changes in carbohydrate metabolism after bariatric/metabolic surgery. Their hormonal secretions before and after bariatric/metabolic surgery have been widely studied in plasma or serum from animals and humans [10,11] but the islet cell composition and its paracrine interactions have been studied less. We will attempt to summarize what we know about the subject by means of a bibliographical review of the most relevant works published on the subject.

## 2. Methods and Results

This paper is a narrative literature review text that aims to expose the framework surrounding the effects of RYGB and SG on endocrine-cell populations in the pancreas. We performed a selective search of numerous articles in different databases, as well as books.

The literature of the main scientific databases was reviewed. The search was limited to documents published between 2001 and 2021. These databases were Medline, PubMed, Chochrane and Scopus. In addition, a search was carried out on academic websites, such as Google Scholar, SciELO and Dialnet. The main Boolean operators used were: AND, OR and NOT, and the key words were sleeve gastrectomy; roux-en-Y gastric bypass; beta-cell, alpha-cell; epsilon-cell; islet; trans-differentiation. Due to the large number of studies found, the following criteria were applied to filter the results and work with the most relevant studies.

Inclusion criteria: Original articles, systematic reviews and meta-analyses concerning modifications of the endocrine pancreas after bariatric or metabolic surgery in humans or animal models. Papers published in English in the last 20 years (2001–2021). We prioritised information from systematic reviews and meta-analyses with high scientific evidence.

Exclusion criteria: Papers not related to the topic or not meeting the inclusion criteria.

In the end, a total of 435 articles were found that met the search criteria. Of these, 47 were selected for the preparation of this manuscript. As Table 1 shows, a large number of disciplines are involved in the study of the topic.

## 3. Discussion

### 3.1. The Sleeve Gastrectomy and the Islet Architecture

Bariatric/metabolic surgery involves different techniques leading to different effects on pancreatic cell populations. Currently, sleeve gastrectomy (SG) is one of the most performed techniques. A consequence of this procedure is the drastic removal of the gastric fundus and corpus ghrelin-producing cell population. This situation leads to 35–45% reduction of blood ghrelin levels after gastrectomy in humans [12,13,14]. However, a recent study described the expansion of the pancreatic residual postnatal epsilon-cell population with recovery of plasma ghrelin levels in rats twelve weeks after SG. This expansion takes place at the expense of pancreatic cell progenitors that differentiate into epsilon-cells showing a high expression of lineage markers such as neurogenin-3 (Ngn-3) but not homeodomain protein Nkx2.2 (Figure 2) [15].

This leads us to believe in an adaptive response of the endocrine pancreas to low circulating ghrelin levels and in a possible explanation of the improvement of beta cell function after SG if we take into account the protective role of ghrelin on it [16].

Furthermore, this surgery does not only affect the epsilon-cells in the islets. It is clear that SG preserves the beta-cell function, at least for a while [17,18]. This could be explained by the increase of GLP-1 receptor expression in beta cells after SG, implying an increase in paracrine sensitivity to GLP-1 [19,20]. However, there are doubts about this due to a recent study with a modified mouse model involving an inducible knockdown of GLP-1r in beta-cells (GLP1rβ-cell-ko), which showed improved glycemic profiles, to the wild-nature level, after SG [21]. Other researchers have linked the maintenance of beta-cell mass and beta-cell identity markers such as PDX-1 or MafA [22] (Picture 2) to high levels of gastrin after SG, as well as to correction of long-term blood glucose levels in rodents [23].

This brings us to the problem of diabetes relapse after SG, which is as high as 41.6% of cases five years after surgery [2]. Liu et al. proposed long-term recovery of insulin sensitivity without beta-cell dysfunction as an answer to the question [24], but a recent work showed loss of beta-cell mass and a strong increase in alpha-cell mass in Wistar rats twelve weeks after SG. Trans-differentiation of the beta-cell population under stressful situations with loss of beta-cell markers such as PDX-1 and gain of alpha-cell markers such as Pax-6 and Arx has been shown [25] (Figure 2). Moreover, this is supported by studies performed on mice outside the scope of bariatric surgery where alpha-cell populations labeled with Gcg-Cre lineage tracers showed a dilution of the marker at the expense of the beta-cell population throughout life [26]. Therefore, the appearance of alpha-cells at the expense of the beta-cell population may explain the long-term relapses in diabetes after SG.

Finally, the protective effect of the somatostatin-14 isoform on Min6 pancreatic beta cells of mice has recently been verified, limiting the stress markers HSPa1 and Ddit3 and apoptosis [27]. This together with the occurrence of delta-cell hyperplasia in Goto-Kakizaki diabetic mice [28] makes us think about a possible role of this delta population in the mechanisms underlying SG. This seems to be reinforced by the ability of ghrelin to activate the paracrine secretion of somatostatin [29] as mentioned above. However, due to the difficulty in carrying out these studies in humans and the ethical aspects, further investigation on animal models is needed to clarify this issue and the possible involvement of other pancreatic endocrine populations.

### 3.2. The Roux-en-Y Gastric Bypass and the Islet Architecture

Roux-en-Y gastric bypass appears to be the most powerful tool for the management of obesity and hyperglycemia in patients [30]. This procedure has demonstrated its efficiency in increasing beta-cell function in animal models and patients [31,32]. It also appears to increase beta-cell mass after surgery in both animal models and patients [33,34]. GLP-1 activity has been proposed as responsible for these effects on beta-cell mass after RYGB [35]. On the other hand, glucose improvement after RYGB has long been reported in mice models of functional GLP-1 and GLP-1 receptor deficiency, suggesting a GLP-1 independent mechanism for glycemic control after surgery [36]. Another very interesting candidate is intra-islet PYY. Guida et al. reported a large increase in islet PYY content after RYGB, mediated by locally produced PYY but not GLP-1 glucose-stimulated insulin secretion. Furthermore, interleukin-22 (IL-22) seems to play a key role in the increase of intra-islet expression of PYY after RYGB. This situation would imply that non-surgical treatment for diabetes is possible [37].

An interesting study would be to determine the participation of pancreatic delta-cells in the maintenance of beta-cell mass after RYGB surgery since a recent study demonstrated that delta-cells become insulin-expressing cells after the ablation of insulin-secreting beta-cells in human islets [38] (Figure 2). This should be investigated in the future.

Other cell types, such as pancreatic epsilon-cells, do not seem to be affected after RYGB [15]. However, high plasma ghrelin levels were detected in obese mice six weeks after RYGB, probably due to an expansion of ghrelin-producing cells in the duodenum and stomach of these mice [39].

On the contrary, the plasticity of the pancreatic alpha-cell population under stressful circumstances is well known. Pregnancy or intermittent fasting are capable of enhancing the alpha-cell mass in mice [40,41]. Some factors related to the functionality of hepatic glucagon receptors (GCgr) have been proposed as brakes and regulators of alpha-cell population expansion in animal models [42]. In this sense, RYGB is also able to cause an increase in the alpha-cell population in mice six months after the operation, including a loss of beta identity markers such as PDX-1 and a gain of alpha-cell markers such as ARX in the islets (Figure 2). All of this suggests long-term trans-differentiation of beta-cells into alpha-cells after surgery [25].

This brings us to long-term relapse of diabetes again. Like SG, the outcomes of RYGB published in relevant trials have shown a progressive worsening of diabetes-related parameters such as glycated hemoglobin, reaching a 50% relapse in diabetes at five years [2]. Patel et al. proposed weak beta-cell function and peripheral insulin resistance as possible causes of relapse after RYGB [43]. An decrease in beta-cell mass and an increase in alpha-cell mass could explain this, but what is the mechanism that triggers trans-differentiation? Hyperinsulinism and subsequent hypoglycemia have been a problem after RYGB but also may be the answer [44]. In this sense, RYGB seems to cause an extreme requirement and stressful situation to the beta-cell population, triggering conversion to alpha-cells [45]. According to this, a study in patients reported hyperinsulinism but elevated postprandial glucagon secretion after RYGB. However, the same study did not report extremely increased beta cell function [46]. The landscape is complex and exciting and could be a good line of research to improve the efficiency of these surgeries in the remission of diabetes.

## 4. Conclusions

SG and RYGB are a therapeutic option not only for overweight but also for diabetes. The effects of these surgeries on enterohormonal levels have been extensively studied but on another level, further research on endocrine pancreatic cell populations is also needed. Nevertheless, it seems that different pathophysiological mechanisms underlie each of these surgeries, at least in reference to their pancreatic involvement. This is a complicated issue in humans. However, a better understanding of the mechanisms and cellular dynamics governing these populations after these two surgeries would allow us to limit hypoglycemic episodes, the relapse of diabetes over time or even the development of pharmacological alternatives to the use of bariatric/metabolic surgery.

## Figures and Tables

**Figure 1 jcm-10-04217-f001:**
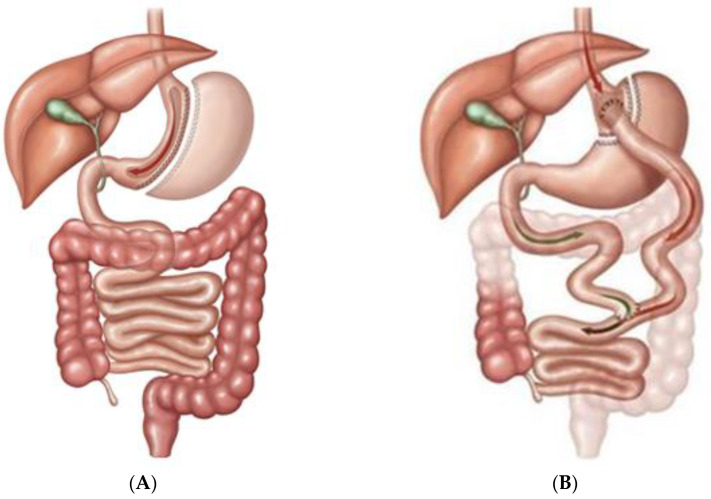
Schematic drawing of Sleeve Gastrectomy and Roux-en-Y Gastric bypass. (**A**) Sleeve Gastrectomy (SG). Representation of a common human sleeve gastrectomy (SG) procedure. The SG is a surgical procedure including a reduction of final gastric volume, since most of the gastric major curvature is resected. The stomach is reduced to a cylindrical pouch removing most of the fundus, stomach-corpus and antrum. The pylorus and minor curvature is preserved. SG reduces the initial stomach volume by approximately 15–20%. In animal models this configuration is maintained since the final gastric pouch volume and valves are preserved. (**B**) Roux-en-Y Gastric Bypass (RYGB). Representation of a common human roux-en-Y gastric bypass (RYGB) surgery. This includes a transverse section of the stomach performed from the major to the minor curvature, configuring a gastric pouch. This pouch of the stomach continues to the food handle with an alimentary bulb, which continues with the medium portion of the jejunum. RYGB, a mixed malabsorptive and restrictive technique, excludes the antrum and the proximal intestine to aliments by bypassing the duodenum and the initial part of the jejunum. This includes biliopancreatic secretion, which determines the malabsorptive component. The biliopancreatic bulb connects with the mid jejunum. In rats, the model was reproduced similarly with minor modifications according to the animal anatomy. Exempli gratia, the jejunal alimentary bulb was 10 cm due to the usual intestinal medium extension of 80 cm. Original figure seen in https://sagebariatric.com/about-surgery-home/sleeve-gastrectomy (accessed on 22 July 2021).

**Figure 2 jcm-10-04217-f002:**
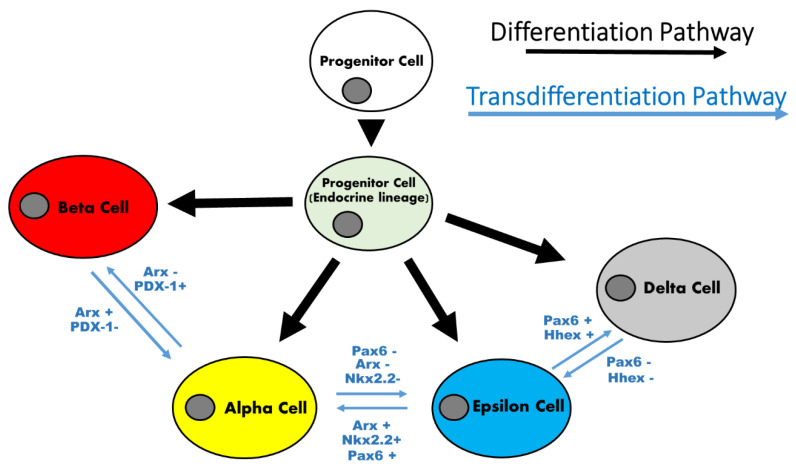
Pancreatic endocrine cell identity markers and possible cell trans/differentiation pathways after SG/RYGB. Pancreatic endocrine-cell identity markers and possible cell differentiation pathways from progenitor-cells (Black arrows) or trans-differentiation from other pancreatic endocrine-cells (Blue arrows) after sleeve gastrectomy or roux-en-Y gastric bypass.

**Table 1 jcm-10-04217-t001:** Search Results. Break down of the total number of articles used to prepare the work. The left column represents the different fields of research of each journal citation (Journal Citation Report categories). The central column contains the number of citations found in each category and the right column contains the number and percentage of citations selected for the manuscript.

Research Field (JCR)	Number of Articles Obtained	Number and % of Articles Selected
Endocrinology & Metabolism	223	22 (46.80%)
Surgery	91	6 (12.76%)
Cell Biology	29	4 (8.50%)
Medicine General & Internal	27	4 (8.50%)
Biochemistry & Molecular Biology	21	2 (4.25%)
Multidisciplinary Sciences	14	2 (4.25%)
Medical Research & Experimental	11	2 (4.25%)
Gastroenterology & Hepatology	11	2 (4.25%)
Genetics & Heredity	4	1 (2.12%)
Pediatrics	3	1 (2.12%)
Peripheral Vascular disease	1	1 (2.12%)
Total of Research fields	435	47 (100%)

## Data Availability

The scientific articles consulted for the preparation of this review were obtained from the following databases: Pubmed, Science Direct, Google Academic and Rodin: (UCA Institutional Repository): It is a database of teaching and research objects of the University of Cadiz.

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
