# Peer review of "Sleeve Gastrectomy and Roux-En-Y Gastric Bypass. Two Sculptors of the Pancreatic Islet"

_jcm, 2021, doi:10.3390/jcm10184217_

Round 1
Reviewer 1 Report
Dear collegues,
First of all thank you for the opportunity to review this very interesting work. The understanding of the metabolic mechanisms of bariatric surgical procedures is the most important part of this clinical research field. With this your work seems to be an extremely important contribution to the behavior of pancreatic islet after bariatric surgery and it needs to be published and recognized in the community.
My biggest concern on your study is the design of it. Ciatation Line 56: "We will attempt to summarize what we know about the subject." So what is the exact approach to the topic? Usually it is a meta-analysis of data or literature research, so we do need to display it like that.
So please mention how did you filter your studies and which parameter did you use to figure out the relevant studies and to take out the irrelevant ones and those with poor design. This part needs to be described in a chapter "methods and results". Please also mention how many overall studies did you recognize and how many of them did you use for your summarizing work.
Line 39: " The RYGB, - mixed malabsorptive and restrictive- bariatric surgery" : This wording is out of date.
Figure 2: Too many abbreviations and shortcuts used, which are not usual for the read and heart to understand. Please create a more simple and clear figure.
Line 105/106: What kind up studies are exactly needed to clear this? Clinical studies on patients or animal experiment?
Line 138/139: This numbers about diabetes relapse are from small trials and critical to use in general. Please mention relevant trials like SM-BOSS or Schauer PR, Bhatt DL, Kirwan JP, Wolski K, Aminian A, Brethauer SA, et al. Bariatric Surgery versus Intensive Medical Therapy for Diabetes-5-Year Outcomes. N Engl J Med. Waltham: Massachusetts Medical Soc; 2017;376:641–51.
Line 143: Citation of non-published data is absolutely critical. Do not use it.
As an example for the structure of such kind of study, I would recommend to observe following work: Bile acids and bariatric surgery; Vance L. Albaugh a, Babak Banan a, Hana Ajouz b, Naji N. Abumrad a, Charles R. Flynn a, *
Thanks once again!
Best wishes!
Author Response
Dear colleague.
We would like to thank you for your comments regarding the manuscript. They have been of great value to us and have helped us to improve it without any doubt.
Point 1. My biggest concern about your study is the design of it. Citation Line 56: "We will attempt to summarize what we know about the subject." So what is the exact approach to the topic? Usually it is a meta-analysis of data or literature research, so we do need to display it like that.
So please mention how did you filter your studies and which parameter did you use to figure out the relevant studies and to take out the irrelevant ones and those with poor design. This part needs to be described in a chapter "methods and results". Please also mention how many overall studies did you recognize and how many of them did you use for your summarizing work.
Response 1: As you have suggested, we have incorporated a section entitled material and methods in which we describe the type of study that is carried out and the criteria used and filters when selecting the bibliography. (Please see the attachment)
Point 2. Line 39: " The RYGB, - mixed malabsorptive and restrictive- bariatric surgery" : This wording is out of date.
Response 2: I indeed, since that terminology is out of date, we have decided to remove this terminology from the text (Please see the attachment)
Point 3. Figure 2: Too many abbreviations and shortcuts used, which are not usual for the read and heart to understand. Please create a more simple and clear figure.
Response 3: The previous figure in an attempt to provide the maximum information appears heavily loaded with acronyms and abbreviations. As suggested, we have limited the number of these to the essential ones and we have indicated the differentiation and transdifferentiation pathways (Please see the attachment)
Point 4. Line 105/106: What kind of studies are exactly needed to clear this? Clinical studies on patients or animal experiments?
Response 4: As indicated, we have incorporated into the text the type of studies that we believe necessary. In this case in animal models given the ethical problems to carry them out in humans (Please see the attachment)
Point 5. Line 138/139: These numbers about diabetes relapse are from small trials and critical to use in general. Please mention relevant trials like SM-BOSS or Schauer PR, Bhatt DL, Kirwan JP, Wolski K, Aminian A, Brethauer SA, et al. Bariatric Surgery versus Intensive Medical Therapy for Diabetes-5-Year Outcomes. N Engl J Med. Waltham: Massachusetts Medical Soc; 2017;376:641–51.
Response 5: We have incorporated more powerful and relevant clinical studies into the work. As suggested: Schauer et al. When we mentioned relapse in diabetes (Please see the attachment)
Point 6. Line 143: Citation of non-published data is absolutely critical. Do not use it.
Response 6: As you suggest we are removed the citation of non-published data (Please see the attachment)
Additionally, as you suggest, an extensive edition of the English language has been made, we attach the corresponding certificate of revision. (Please see the link) https://drive.google.com/file/d/1czE6yuhmb8wM04fgauKrdW1NJOd4lwo3/view?usp=sharing
Best regards.

Reviewer 2 Report
Dear Authors.
As I myself am very involved with diabetes remission after bariatric surgery and dumping syndrome in particular, I find your work very interesting. However, there is still clear potential for improvement. For example, the operations are not described correctly.
Roux-en-Y gastric bypass:
The gastric pouch is performed by stapler dissection 5 cm below the esophageal-gastric junction with one cartrage and then upwarts to the angle of HIs. The alimentary limb (adjunct to the gastric pouch) is 150 cm of length in the classic Roux-en-Y gastric bypass. The biliopancreatic limb is 60 cm. These length have been changes, and nowadys moct surgeons perform it vice versa, means a long BP-limb, and a shorter alimentary limb. There is no anastomisis 100-150 cm before the Bauhini valve!
Sleeve gastrectomy:
The antrum is not or only slightly touched. Thats why the gastric pump remains. It is a vertical resection of excess corpus volume, callibrated with a 36-42 French tube, and a complete resection of the fundus with a 1 cm safety margin towards the hiatus. It ramains a high-pressure systems: a small sleeve with high intralumnal pressure from the nearly untouched antrum.
Regarding your thesis:
It is interessing as I mentioned above, but you did not mention with one word that weight loss itself is a generell motor of diabetes remission. Additionally the patients do not eat solid food for 4 weeks after the surgery! The remission with sleeve gastrectomy is much slower as with gastric bypass, which has obviously a direct metabolic effect (look for results with Endobarrier (duodeno-jejunal bypass liner) or duodenal resurfacing as two vivo models of duodenal exclusion).
This is why I think you should revise your manuscript thoroughly. It contains indeed a very interessing thesis, and it is worth to be reworked.
Author Response
Dear Referee
Thanks so much for your considerations and arguments. We accept your constructive comments and appreciate these.
Firstly, the manuscript was revised throughtly by American Journal Experts, an especialized company in the spealing correction of manuscript. We can include AJE certificate. Please see link :
https://drive.google.com/file/d/1czE6yuhmb8wM04fgauKrdW1NJOd4lwo3/view?usp=sharing
We found two aspect according your suggestions. One for surgical procedures and the othe one about functional consequences after these bariatric surgeries.
Points1 and 2: The operations are not described correctly.
Response 1 and 2: About surgeries, the whole detailed procedure was reported in the refered bibliography. Many of the observed data were collected of previous surgical experiences underwent in animal. The precise aspect about the surgeries were included in these referenced reports. All procedures and data are related to rats, the animals selected for experiences. Thus, the similar considerations to human clinic has been confirmed not only in our experiences but in other similar published reports. We review the manuscript in order to correct any misundertood idea of the surgical procedures. We´ll try to clarify these aspects.
In RYGB, we reached an 80% survival goal after surgery in rats. The gastric pouch included the rumen portion in rats. This is an accepted condition in common literature. In the initial surgeries, we probed with different jejunal segment. It was finally limited to 10 cm of alimentary tract. The Bauhin valve was conserved, meanwhile the whole rat intestinal tract is over 80 cm.
About SG, may be we did not express clearly. We´ll rewrite this. Our procedure reproduce the human technique. The gastric volumen was reduced to the final 20% of initial capacity. Since the His angle to antrum, we resected the main fundus and corpus following a parallel line to minor curvature, which was preserved. To this, we used endoluminal esophagueal guides or curved unharmful forceps.
As we wrote in this comment, we rewrite some aspects of the text which can be confuse to the readers, about surgical procedures. We promote the following changes in the text about the surgical technique, in order to specify the main aspects of them:
Figure 1. Figure 1A. It represents a common human Sleeve Gastrectomy (SG) procedure. The SG is a surgical procedure including a reduction of final gastric volume, since most of the gastric major curvature is resected. Gastric is reduced to a cylindrical pouch removing most of the fundus, stomach-corpus and antrum. The pylorus and minor curvature is preserved. The SG reduced the initial stomach volume by approximately 15-20%. In animal models this configuration is maintained since the final gastric pouch volume and valves are preserved. Figure 1B. This represents a common human Roux-en-Y Gastric Bypass surgery (RYGB). This includes a transverse section of the stomach performed from the major to the minor curvature, configuring a gastric pouch. This pouch of the stomach continues to the food handle with an alimentary bulb which continues with the medium portion of jejunum. The RYGB -mixed malabsorptive and restrictive technique- excluded the antrum and the proximal intestine to aliments by the bypass of the duodenum and the initial part of the jejunum. This included the biliopancreatic secretion that determines the malabsorptive component. The biliopancreatic bulb connects with medium jejunum. In rats the model was reproduced similarly with minor modifications according to the animal anatomy. Exempli gratia, jejunal alimentary bulb was 10 cm due to the usual intestinal medium extension of 80 cm. (Please see the attachment)
Point 3: Regarding your thesis:
It is interessing as I mentioned above, but you did not mention with one word that weight loss itself is a generell motor of diabetes remission. Additionally the patients do not eat solid food for 4 weeks after the surgery! The remission with sleeve gastrectomy is much slower as with gastric bypass, which has obviously a direct metabolic effect (look for results with Endobarrier (duodeno-jejunal bypass liner) or duodenal resurfacing as two vivo models of duodenal exclusion).
Response 3: After surgeries, animals suffered a liquid diet period of 72 h as decribed in bibliography. We reported weght loss and the importance in glycemic control. But these consideration were related to human. Certainly, we agree there is an initial al long-term consequence linked to weight loss. But this cannot explain the improvement T2DM, as other metabolic goals. Not only weight loss, but enterohormones serum levels and pancreas cellularity seems to be in the basis of this improvement. So, our manuscript is involved in the animal expression of these endocrine pancreas cellular changes and, probably, the basis not only of the metabolic initial positive consequences; but the late final causes that could be in the basis too of these patients who after 10 years of surgical treatment became newly diabetic.
Additionaly for a better understanding and organization of the work we have incorporated a section entitled: Methods and results in wich we describe the type of study that is carried out and the criteria used and filters when selecting the bibliogrtaphy (please see the attachment)
Best regards

Reviewer 3 Report
Please see comments for Editors
Author Response
Dear Referee,
The manuscript was corrected for American Journal Expert, which certificate can be observed in the next link:
https://drive.google.com/file/d/1czE6yuhmb8wM04fgauKrdW1NJOd4lwo3/view?usp=sharing
Additionally, for a better understanding and organization of the work, we have incorporated a section entitled methods and results in which we describe the type of study that is carried out and the criteria used and filters when selecting the bibliography. (Please see the attachment)
Unfortunately, we cannot contribute with other arguments because the short comment we can read.
Best regards

Round 2
Reviewer 1 Report
The part "methods and results" is much better now. The methods are now well described, but the concrete results are still not tangible. Please make a figure or so to show what kind of papers did you include (for example: anatomical research, physiology, rodent experiments, clinical studi9es? etc.) and also please display the results in understandable way. The discussion is fine, but a base of results is still missing.
Author Response
Dear colleage
As you suggest, we have organized the results obtained after the bibliographic search in a table. We have broken down the results according to the different fields of Knowledge and indicated the percentage of these in the study. (please see the attachment)
Thank you again.
Best regards.
